# Investigating Cooking Activity Patterns and Perceptions of Air Quality Interventions among Women in Urban Rwanda

**DOI:** 10.3390/ijerph18115984

**Published:** 2021-06-02

**Authors:** Catherine A. Campbell, Suzanne E. Bartington, Katherine E. Woolley, Francis D. Pope, Graham Neil Thomas, Ajit Singh, William R. Avis, Patrick R. Tumwizere, Clement Uwanyirigira, Pacifique Abimana, Telesphore Kabera

**Affiliations:** 1College of Medical and Dental Sciences, University of Birmingham, Edgbaston, Birmingham B15 2TT, UK; cac434@alumni.bham.ac.uk; 2NHS Lothian, Waverly Gate, 2-4 Waterloo Place, Edinburgh EH1 3EG, UK; 3Institute of Applied Health Research, University of Birmingham, Edgbaston, Birmingham B15 2TT, UK; kew863@student.bham.ac.uk (K.E.W.); g.n.thomas@bham.ac.uk (G.N.T.); 4School of Geography, Earth and Environmental Sciences, University of Birmingham, Edgbaston, Birmingham B15 2TT, UK; f.pope@bham.ac.uk (F.D.P.); a.singh.2@bham.ac.uk (A.S.); 5International Development, School of Government, University of Birmingham, Edgbaston, Birmingham B15 2TT, UK; w.r.avis@bham.ac.uk; 6College of Science and Technology, University of Rwanda, Avenue de l’Armee, Kigali P.O. Box 3900, Rwanda; remick05@yahoo.fr (P.R.T.); uwanyirigiraclement@gmail.com (C.U.); pgodgiven@yahoo.com (P.A.); kaberacris@yahoo.fr (T.K.)

**Keywords:** household air pollution, biomass fuel, charcoal, air quality interventions, improved cookstove, cooking activities, urban, Rwanda, women

## Abstract

Household air pollution (HAP) from biomass cooking with traditional stoves is a major cause of morbidity and mortality in low-and-middle-income countries (LMICs) worldwide. Air quality interventions such as improved cookstoves (ICS) may mitigate HAP-related impacts; however, poor understanding of contextual socio-cultural factors such as local cooking practices have limited their widespread adoption. Policymakers and stakeholders require an understanding of local cooking practices to inform effective HAP interventions which meet end-user needs. A semi-structured questionnaire was administered to 36 women residing in biomass-cooking fuel households in Kigali, Rwanda to identify cooking activity patterns, awareness of HAP-related health risks and ICS intervention preferences. Overall, 94% of respondents exclusively used charcoal cooking fuel and 53% cooked one meal each day (range = 1–3 meals). Women were significantly more likely to cook outdoors compared to indoors (64% vs. 36%; *p* < 0.05). Over half of respondents (53%) were unaware of HAP-related health risks and 64% had no prior awareness of ICS. Participants expressed preferences for stove mobility (89%) and facility for multiple pans (53%) within an ICS intervention. Our findings highlight the need for HAP interventions to be flexible to suit a range of cooking patterns and preferred features for end-users in this context.

## 1. Introduction

The combustion of solid biomass fuels (wood, charcoal and dung) for cooking, lighting and heating is the primary source of household air pollution (HAP) in developing countries [1]. The combustion of solid fuels using traditional stoves typically produces pollutant levels which exceed World Health Organization (WHO) indoor air quality guideline levels [2]. Although evidence remains limited, there are approximately four million premature deaths associated with HAP exposure worldwide each year [3], a burden which is greater than any other major modifiable risk factor in Africa, including unsafe water and child malnutrition [4]. Known adverse health events associated with HAP exposure include respiratory tract infections [5,6], chronic obstructive pulmonary disease [7], lung cancer [8], ischaemic heart disease [9], adverse pregnancy outcomes [10], cognitive impairment or decline [11,12], cataracts [13] and burn injuries [6]. HAP exposure is often associated with social inequalities, with women and young children being disproportionately adversely affected in terms of health (due to increased exposure), education and income-generating opportunities, dedicating many daily hours to cooking and fuel gathering tasks [14,15,16]. In addition, the environmental burden from unsustainable harvesting and burning of biomass fuels contributes to carbon emissions, climate change, deforestation and flooding [17].

Implementation of different harm reduction (e.g., improved ventilation, behaviour change, educational initiatives and improved cookstoves [ICS]) and harm prevention fuel transition interventions (e.g., Liquid Petroleum gas [LPG], electricity and solar), to combat the public health and environmental burden of HAP have typically had limited success in Sub-Saharan Africa [18,19,20]. Challenges for cleaner cooking intervention development and implementation include the limited consideration for household needs and cultural preferences [18], unreliable access and financial constraints. These factors may contribute to household ‘stove stacking’ [21] whereby women continue to use traditional stoves alongside ICS, thus, retaining traditional modes of cooking [22]. Research across East Africa has highlighted the past failure of policy makers to understand issues around affordability, acceptability and accessibility of solutions [23]. There is also frequently a lack of perceived need for a HAP intervention [12,24], with reduced uptake among those vulnerable groups with a lower level of education [25,26], therefore implementation risks widening existing health and social inequalities. Furthermore, distinct differences exist between communities in their cooking activities, health knowledge and attitudes to cleaner cooking technology [24,25]. Therefore, initiatives must consider local contextual and compositional factors within the target community prior to intervention design and implementation [27]. Overall, experience of HAP harm mitigation interventions in low-and-middle-income country (LMIC) settings has indicated there are economic, cultural and educational challenges to successful long-term adoption [18,19,20].

Biomass fuels are used by 98.1% of households in Rwanda [28], East Africa, a country which has undergone a rapid expansion and urbanisation of its population in recent years, resulting in a high population density (498.7 people per km^2^) [29]. Rwanda’s economy has also rapidly expanded in recent decades, with a Gross Domestic Product (GDP) of $773 per capita [30] and an ambitious aim to become an upper-middle income country by 2035 [31]. Kigali, the capital and main urban area of Rwanda has a population of 1,132,686 inhabitants [32], distributed across three districts (Gasabo, Kicukiro and Nyarugenge), subdivided into sectors, cells, then villages. The primary domestic cooking fuel is charcoal, used by 66% of households in the city while the remaining households use wood (26%), straw/grass (3%) or LPG (2%) [28]. Current HAP interventions in Rwanda include introduction of LPG (HAPIN randomised controlled trial) [33], biomass pellets [34,35], modifed stoves ICSs [36]; however, there remains a paucity of evidence to inform effective real-world HAP mitigation strategies within the local setting.

Therefore, this cross-sectional study focuses on women living in biomass fuel households in urban Kigali and aims to: (i) identify local cooking activity patterns, (ii) explore awareness of the health risks associated with HAP exposure and (iii) investigate user preferences for air quality interventions. The data presented are valuable for informing the development of future HAP harm prevention and mitigation measures in urban Rwanda, in accordance with best practice for intervention development [27]. 

## 2. Materials and Methods

### 2.1. Study Setting and Participants

This urban-based study was conducted across households within the Muhima Sector, of the Nyarugenge District of Kigali, Rwanda. Field study was undertaken between January and March 2018, within the ‘short’ dry season with average temperatures 17–28 °C.

Of the three districts forming Kigali, Nyarugenge has the highest population density with 2149 inhabitants/km^2^ [27]. The seven villages participating in this study were all from the Kabeza cell within the Muhima Sector. In Muhima, 94% of inhabitants reside in non-permanent dwellings composed of corrugated iron roofing, sun-dried bricks, wood and mud [37] and the majority of households (87%) depend on charcoal fuel for cooking [37].

Seven villages within Kabeza, a cell within Muhima Sector were selected by convenience sampling due to location, type and number of households. Since HAP impacts disproportionately upon women and children [14,15,16], dwellings inhabited by women and young children were the focus of this study. Households were considered eligible for the study if they: (i) exclusively used a biomass fuel cookstove, (ii) had one female resident aged 18–55 years and (iii) had at least one child under five years. A stratified convenience sample of 40 households was selected from 211 households that met the eligibility criteria. The sample was stratified by village, with the number of households for each village unit determined proportionately based upon the number of eligible households.

### 2.2. Study Design and Data Collection

This study employed a semi-structured questionnaire (Appendix A) administered by trained local fieldworkers to adult women with responsibility for domestic cooking within eligible study households. The questionnaire was informed by relevant literature and modelled on core questions within validated questionnaires produced by the WHO [38] and the World Bank [39]. Prior to deployment, the questionnaire was piloted among three household respondents, with modifications to question length and terminology. Questions were asked in Kinyarwanda, the local language, with responses translated verbally and recorded in English by trained fieldworkers. If there was difficulty in comprehending questions among respondents, they were amended and re-checked. The final study questionnaire comprised 26 open and closed questions (in English) across three sections: (i) cooking activities (i.e., fuel and stove choice, number of meals cooked per day and time spent cooking), (ii) awareness of the health risks associated with biomass cooking smoke and (iii) perceptions and preferences regarding air quality interventions (i.e., ICS, gas fuel and improving household ventilation), with an overall completion time of approximately 30 min.

### 2.3. Data Analysis

Questionnaire results were quantified, and statistical analysis was performed using IBM SPSS version 24.0 (IBM Corp, Armonk, NY, USA, 2017) [40] and Stata version 16 (StataCorp LLC, TX, USA, 2019) [41]. Descriptive analyses were undertaken to provide summary statistics (frequencies, percentages, medians and interquartile ranges) for outcome variables. Univariate analyses were used to investigate the three primary outcomes: (i) local cooking activity patterns, (ii) awareness of the health risks associated with cooking smoke and (iii) perceptions and preferences regarding air quality interventions.

### 2.4. Ethical Approval

This study received local ethical approval from the University of Rwanda, College of Medicine and Health Sciences Institutional Review Board (No. 317 CMHS IRB) and the University of Birmingham, Internal Research Ethics Committee (IREC2017/1413634). Fully informed, written consent was obtained from each study participant and all respondents were free to withdraw from the study at any time.

## 3. Results

The overall participation rate was 100% with all women approached willing to partake in the study. In total, 36 respondents who met all inclusion criteria completed the questionnaire and were included in the final data analysis. The median age of participants was 31 years (range = 20–49 years) and in 94% of households the mother was the main household cook (Table 1). Most households had one child aged under five years (69%). More than half of households (64%) had between five to ten occupants (older children and grandparents). No dwelling had a chimney and whilst 100% of participants reported to have household electricity access, this study did not collect data on whether the households paid for the supply of this utility, or how frequently it was used.

### 3.1. Cooking Activity Patterns

Fuel and stove choice were consistent cooking characteristics among study households. All 36 participants used charcoal as their primary cooking fuel with only two reporting the occasional use of wood. 92% of women reported using charcoal because it was the cheapest fuel (Table 2). In 81% of households, plastic was used to aid lighting the stove–this was often broken fragments of water containers or food packaging. A mobile single-pot metal charcoal stove was used by all but one (97%) of the women who used a fixed multi-pot stone stove built into her kitchen area. 

The most common stove type was consistent in design, with the core structure comprising an outer body of metal cladding, an interior ceramic grate where charcoal is placed and an air vent opening at the base (Figure 1a–c). 17% of women reported owning multiple single-pot metal charcoal stoves. Cooking activities varied among households, particularly for the stove location, the number of meals cooked per day and time spent near the stove. Overall, women were significantly more likely to cook outdoors compared to indoors (64% vs. 36%; *p* < 0.05; Table 2). Sixty-four percent (64%) of women reported cooking outdoors in their yards, often on ledges in close proximity to their front doors (Figure 2a) with all but one reporting to cook inside when it rained. Nine women (25%) had a separate kitchen area, which was a shelter either attached to (Figure 2b) or separate from the main house (Figure 2c).

Of 35 women who provided information about cooking times, just over half (51%) cooked one meal per day, most commonly in the morning between 10 a.m. and 12 noon. These women commented that they cooked a large quantity of food at once so that there was enough to eat later in the day. Women cooking two (37%) or three (11%) meals per day also frequently reported cooking between 10 a.m. and 12 noon and 6 p.m. and 8 p.m. The total time women spent attending their cookstove each day ranged from a minimum of 1 h to a maximum of 8 h (median = 3 h), meaning that some women were spending up to one third of every 24 h period near their stove (within 2 m). Women who cooked more meals each day spent longer on average near their stoves, as did their children (Figure 3), with the accompanying exposure to cooking smoke. Women cooking one meal per day spent on average (median) 2 h near their stove whilst women cooking three meals per day spent 5 h on average. For each ‘meal group’, the time the child spent near the stove was less than the women.

### 3.2. Awareness of the Health Risks Associated with Exposure to Cooking Smoke

Over half of participants (53%) were unaware that the smoke from charcoal cookstoves could cause health problems for themselves and other household members. Furthermore, the majority of women (86%) allowed their children near the cookstove (within 2 m). Participants identified several barriers to preventing their child’s close exposure to the stove and hence cooking smoke, which are illustrated in Figure 4. The most common reason cited was the lack of alternative childcare, meaning that the child must accompany them during cooking sessions at the stove. Four of the five households where children were not allowed near the stove reported this was because they feared their child being burnt.

Women participants self-reported a range of symptoms associated with smoke exposure with 81% experiencing eye problems and 67% a cough as well as a runny nose (41%), breathing problems (39%) and itchy skin (8%). For children under five years, the most common symptoms reported by their mothers were a cough (65%) and breathing problems (45%) with less experiencing runny nose (40%), eye problems (40%) and itchy skin (12%). Having been informed about the health risks resulting from exposure to HAP from charcoal cooking smoke, almost all (94%) of respondents reported to be concerned for their and their family’s health. Additionally, every participant requested more information.

### 3.3. Perceptions of Air Quality Interventions

Only 33% of women had heard of an improved cookstove (ICS) prior to receiving this questionnaire, yet all participants reported that they would be interested to try one. Women were asked on their design preferences for an ICS; these are illustrated in Figure 5 and show the most desired features were for it to be mobile (89%) and to have space for more than one pan (53%).

Table 3 summarises participants’ perceptions about the use of gas fuelled stoves versus traditional charcoal cookstoves. Overall, 81% of women were interested in trying gas fuel. The most common reasons cited for this were reduced cooking time (56%), release of less/no smoke (44%), cleaner cooking fuel (25%) and easier to use and ignite (25%). However, 44% of women expressed concern about the cost of gas, listing this as a major barrier to its long-term adoption. Women were also anxious about the safety of using gas around young children (25%) and worried about the risk of fires and explosions (25%). Just under one fifth of women (19%) cited that they were not interested in using gas at all.

Commonly reported negative experiences of cooking with traditional charcoal stoves include associated health problems from cooking smoke (28%), the fact that the stoves break easily (19%), charcoal is dirty to cook with (17%) and makes food and water taste of smoke (14%).

Women were also questioned about household ventilation adjustments with 36% being receptive to changing the stove position and 42% considering installation of a chimney, while only 11% reported they would consider additional windows/vents. The largest barrier to accepting household ventilation interventions was not being a homeowner (33%), other barriers reported were a lack of space (14%) and the cost (3%).

## 4. Discussion

In this novel cross-sectional study, 36 women from urban households in Kigali, Rwanda, participated in a semi-structured questionnaire, providing information regarding their cooking activities, awareness of HAP-related health risks and perceptions of air quality interventions, with a 100% response rate. The findings show that there is a high level of interest in the adoption of household air quality interventions in this community setting, with the majority of women expressing interest in ICS or gas cooker installation. However, overall awareness of the health risks associated with exposure to cooking smoke was suboptimal, with less than half of mothers (47%) aware of any of the health consequences of charcoal smoke exposure. This paucity of knowledge suggests there is potential for an educational initiative alongside a structural intervention.

Overall, one quarter of participants regularly cooked in a separate kitchen area, normally a small building or room either attached or detached from the house. Past studies have used this as a proxy for wealth, which is reflected in the association between increased household wealth and increased likelihood of adopting ICS interventions [25,26]. Socioeconomic factors were also presented as barriers to the adoption of air quality interventions, with 33% not owning their property and 14% reporting a lack of household space. Participants also identified negative factors concerning current cooking practices, including traditional charcoal cookstoves being easily broken (20%) and dirty to cook with (17%), in addition to the health risks including burns and injuries (45%). The improved understanding of local contextual factors identified in this survey is highly valuable for formative development of an effective public health intervention in this local context [27].

### 4.1. Cooking Activity Patterns

Previous studies have illustrated biomass fuel cooking practices to be strongly locally determined, with variation across different contexts [24,25]. In contrast to the practices identified in our current study, research undertaken in rural villages in the Muhanga district (Southern province) and Gakenke district (Northern province) of Rwanda [42] identified that the majority of the households use traditional three-stone firewood stoves (89%) rather than a portable single cookstove (12%). However, similarly householders preferred to cook outdoors (68%) [42]. Rural communities in Kenya used three-stone firewood stoves with seasonality determining their cooking location, mainly inside but outside during hot summer weather [43]. Likewise, in a peri-urban Ugandan study, householders cooked inside using charcoal or wood in clay or three-stone stoves [44]. These data highlight the distinction in cooking practices between rural and urban settings within East Africa, reiterating the importance of collecting and analysing local community level data to inform intervention development.

The present study found that women spent between one and eight hours per day attending to their stoves, meaning that some were spending up to one third of every day in close proximity to charcoal cooking smoke. A further study performed by our group attending the same households to measure levels of particulate matter and carbon monoxide identified concentrations up to four times greater than the WHO air quality standards [45]. This will have obvious health consequences for both women and their young children as a result of significant chronic lifetime exposure. Further, the high proportion of respondents who reported using plastic to light the stove (81%) suggests further research regarding the implications of this practice is merited in this setting.

### 4.2. Awareness of the Health Risks Associated with Exposure to Cooking Smoke

Several previous studies undertaken in LMIC contexts have identified limited awareness of the health risks associated with cooking smoke, with participants aware of short-term consequences of smoke exposure (e.g., coughs and eye irritation) but with little knowledge of long-term health problems [24,44,46]. Our findings are consistent with these studies, with almost all (94%) women reporting to experience smoke-related symptoms; however, less than half reported awareness of the serious health risks associated with HAP exposure. Whilst most women reported they were worried about these health effects, some expressed an acceptance of the situation, given the limited choice in cooking practices and felt that the situation could not be easily changed.

Previous qualitative research undertaken in Peru [47], identified that 19% (*n* = 31) of women did not want to swap from charcoal to gas (LPG) fuel, perceiving it to be more dangerous for household members and expressing a fear of fires, explosions and burn injuries. Similarly, women in our study were concerned about the safety of using gas around young children (25%) as well as the risk of fires and explosions (25%). There is therefore a gap in knowledge concerning the comparative risks of biomass and LPG fuel cooking, with exposure to charcoal cooking smoke recognised as posing a much greater risk to long-term health than the risk of explosions or burns associated with LPG usage [47,48].

The available evidence suggests that knowledge concerning traditional stoves being harmful for health is associated with motivation to purchase an ICS [46]. However, trials of ICS interventions frequently report stove stacking as a challenge, which mitigates the health benefits [25,42]. These findings emphasise the need for educational initiatives alongside structural interventions to achieve maximum impact among the target group in this context. In Uganda and Kenya, the use of radio as an educational platform to inform communities about the health and environmental impacts of using traditional biomass cookstoves has shown to be an important factor in increasing the adoption of ICS [44,46]. In the current study, only one woman reported hearing about the health risks from charcoal cooking smoke on the radio, possibly due to a lack of access to a radio or that it is not a topic regularly discussed on local radio stations. ‘Umuganda’, a monthly nation-wide mandatory community service day whereby members of the public gather for several hours to work on local community projects may be a useful alternative platform to disseminate knowledge of HAP health risks and improved cookstoves. The FRESH AIR project in Uganda [49] piloted a midwife-led health education strategy, teaching 244 pregnant women about the risks of biomass exposure and was found to have positive behavioural impacts on their cooking practices. Mechanisms for effective dissemination of educational materials should be a consideration for future air quality educational interventions in Rwanda.

### 4.3. Perceptions and Acceptability of Air Quality Interventions

Women frequently reported that traditional charcoal stoves emit large volumes of smoke, which they reported to irritate their eyes, nose and lungs. Other common concerns expressed included charcoal being dirty, difficult to light, time consuming to cook with and made food taste smoky. Furthermore, women complained frequently about burn injuries and that the stoves were fragile and easily broken. Overall, these negative perceptions of traditional stoves are shared by women from a variety of settings across the literature [18,19,24,43]. One key distinction noted in previous studies was household preference for the taste of food cooked on traditional stoves compared to ICS or alternatively fuelled stoves, such as LPG [19,44,47]. This contrasts directly with the findings of the present study, which highlighted that the smoky taste of food from traditional charcoal stoves would encourage stove change among households.

Only 33% of women participating were aware of ICS. These findings are consistent with a qualitative study in Uganda where most participants were unable to describe an ICS [44]. Women in the present study received an explanation about ICS and subsequently, all 36 participants expressed an interest in trying one. The emerging stove design priorities for these women were for it to be mobile (89%), have space for multiple pans (53%), cook food faster (11%) and use less charcoal (6%). These findings in part conflict with another study in peri-urban Uganda, where women ranked stove portability and ability to cook multiple dishes at once as less important than cooking food quickly, being affordable and durable [44]. This might be related to the nature of urban living in the current investigation whereby households are more likely to be of rental tenure and thus families move between dwellings more frequently. Therefore, having a mobile stove was a priority for these women. Furthermore, this could also be a reflection on the difference in weather; Kigali has a lower annual average rainfall than Kampala (Uganda) [50,51], which perhaps explains higher rates of outdoor cooking in the current study and thus portablity may be ranked highly to enable the stove to be moved inside when it is raining.

Hence, stakeholders developing ICS initiatives should consider adopting co-development processes, which consider cooking needs and desired attributes to ensure their widespread adoption and use.

Most participants expressed interest in using gas with the top four perceived benefits being (i) gas cooks food faster, (ii) it releases no/less smoke, (iii) it is easy to use and light and (iv) is cleaner and improves sanitation. Women trialling LPG in previous studies reported similar benefits–easier to use, quicker to cook food and significantly cleaner [19,47]. This information together with women’s experiences of traditional biomass stoves provides insight into the factors facilitating and underpinning the process of adoption of air quality interventions.

Mirroring former research [19,47], the risk of fires, burns and gas explosions typically deterred participants from wanting to adopt gas fuel. Additionally, gas was deemed unsuitable for cooking certain time-consuming foods such as beans and cassava. This has also been reported in rural Rwanda [42] and peri-urban Uganda [44]. Households in this study expressed cost as the major barrier to the transition to clean cooking, with the capital expense of gas installation on top of monthly fuel costs being the main concern, a common theme in multiple studies [18,19,26,46]. Whilst a few households (6%) perceived the monthly cost of gas to be cheaper than the cost of charcoal and though this is true [52], it is a well recognised economic phenomenon that individuals place greater emphasis on the upfront capital costs rather than the running costs [53]. Furthermore, this may also reflect the lack of knowledge about alternative domestic fuels and further backs the need for educational public health campaigns to raise awareness. Biomass ICS were considered a more acceptable alternative; however, concerns were expressed about overall costs. Thus, promoting the understanding of fuel economy savings (i.e., shorter cooking times mean reduced fuel consumption as well as more time for women to perform other income-generating activities) may be an important consideration for short and medium term HAP initiatives, with potential to encourage adoption rates.

Finally, women were asked if they would be willing to consider installing a chimney, windows or moving the position of their stove to improve household ventilation. Overall, there was limited enthusiasm for other interventions, with only four women (11%) keen to install additional windows. The primary barrier to such alterations was identified as housing tenure, with most renting the home, thereby restricting ability to make household alterations, an important consideration reflecting the need for engagement with urban residential dwelling owners in this context. Shortage of space and cost of installation were other hindrances. These findings have implications for policy makers when considering socio-cultural barriers to intervention uptake in this community.

### 4.4. Potential Limitations

Our study was cross-sectional and therefore, limited to a single time-point and did not identify changes in cooking patterns or trends over time. The findings of this study have limited generalisability given the small sample setting. However, the aim of this research was to collect data on the local urban community in order to guide air quality interventions that would be targetted to the needs of this specific population, encouraging long-term adoption and subsequently improving public health.

Overall, all study participants (100%) reported a willingness to try an ICS. This may perhaps be attributed to responder bias, whereby despite assurance of research confidentiality, the participant’s hope of receipt of an ICS may have affected their comments.

Finally, the fieldwork interviewers were fluent in English; however, nuances in language may have been lost in the oral translation of participant responses from Kinyarwanda, particularly in the answers to open questions, which were not possible to translate verbatum. Furthermore, it is prudent to consider the impact of influences of a western researcher and individuals not accustomed to the local culture on the interpretation of the women’s responses. The potential for misunderstandings were minimised by working with local fieldworkers and responses were checked to ensure the correct comprehension to remove doubt.

## 5. Conclusions

This study is to our knowledge the first to describe cooking patterns, the awareness of the health risks from exposure to cooking smoke and the perceptions of air quality interventions among women living in urban Kigali, Rwanda. With international pressure to scale-up ICS and alternative HAP intervention programmes, it is crucial that they are end-user focused and understand the local cooking needs, behaviours and barriers.

Overall, our findings are important for the formative development of an effective household air quality harm mitigation intervention to deliver health benefits for women and children in this urban setting. The in-depth, semi-structured questionnaire enabled rich data collection. Engagement with local fieldworkers and community leaders helps ensure the support of the study and supported research capacity building. We gained detailed knowledge of housing characteristics, cooking practices as well as barriers and facilitators to the adoption of domestic air quality interventions.

Our findings are essential to inform the initial development stages for a complex public health intervention for HAP exposure mitigation, in accordance with the Medical Research Council (MRC) Framework for Complex Interventions [27], including recommendations for feasibility and pilot research to provide an understanding of the theories underlying the intervention design, implementation and uptake [27]. We have also documented priority design features of an ICS, thus facilitating the early development in this specific urban community. Importantly, our findings also identify the need for increasing educational awareness of the long-term health impacts associated with HAP exposure, which may be addressed by adoption and evaluation of relevant health promotional approaches. In conclusion, the findings from this study may be used to guide implementation of effective future public health interventions to reduce HAP exposure in this urban community.

## Figures and Tables

**Figure 1 ijerph-18-05984-f001:**
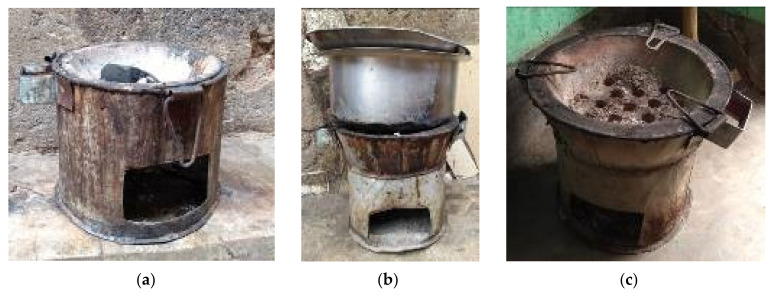
(**a**–**c**). Selected single-pot metal charcoal stoves used in Nyarugenge District of Kigali.

**Figure 2 ijerph-18-05984-f002:**
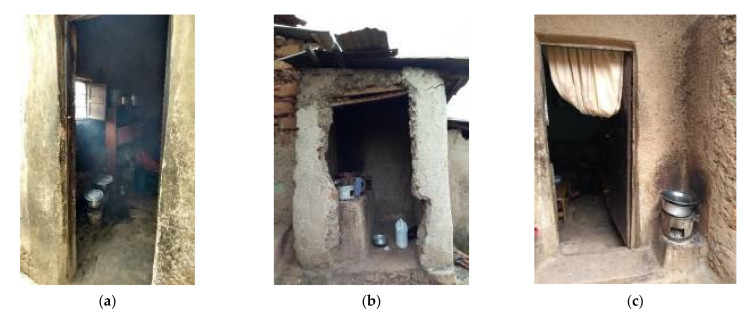
(**a**–**c**). Household cooking areas in Nyarugenge District of Kigali.

**Figure 3 ijerph-18-05984-f003:**
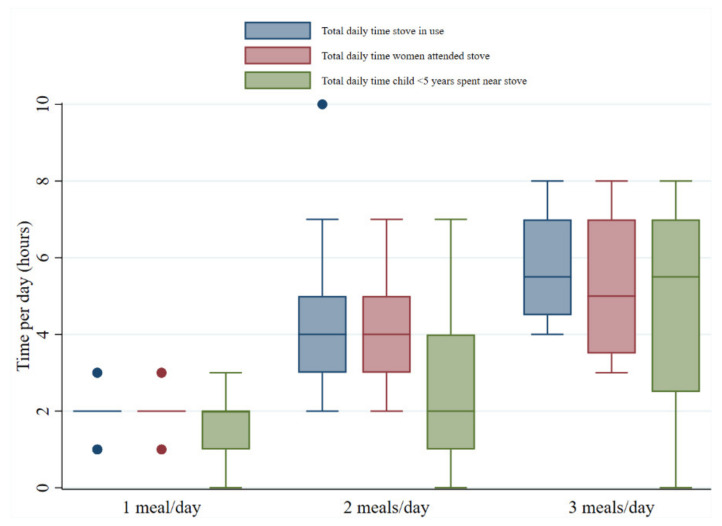
Box and whisker plot showing cooking times and exposure duration.

**Figure 4 ijerph-18-05984-f004:**
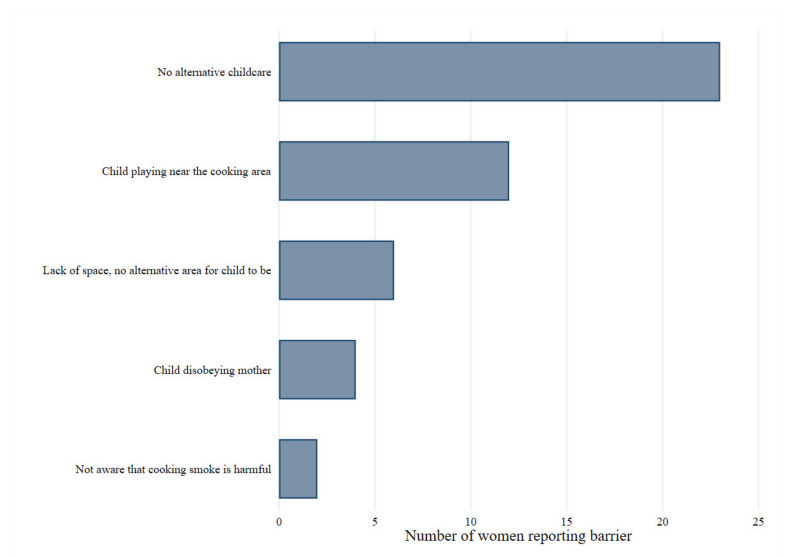
Reported barriers to preventing child’s exposure to the cookstove.

**Figure 5 ijerph-18-05984-f005:**
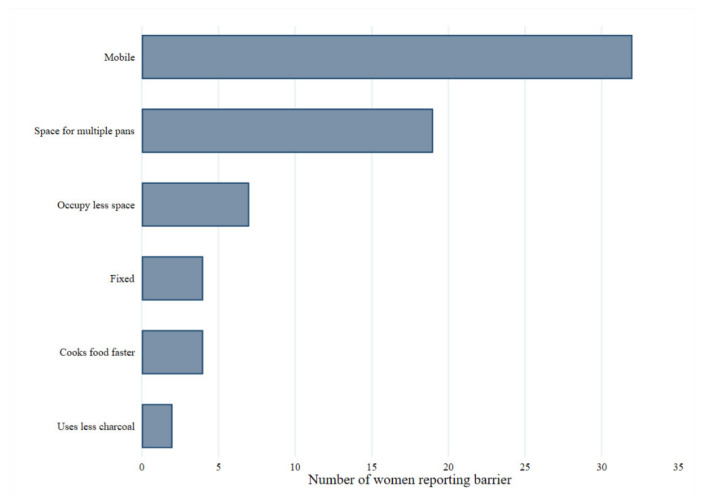
Participant responses for desired features of improved cookstove. Women participating also expressed some concerns regarding ICS, these included cost (14%), the fact that it still released smoke (6%), it would still contribute to health (11%) and safety issues (3%), as well as impact on cooking time for foods (6%).

**Table 1 ijerph-18-05984-t001:** Background demographic and household information (*n* = 36).

Demographic Variables	Study Participants *n* (%)
Village:	
Sangwa	13 (36.1)
Ingenzi	3 (8.3)
Ituze	7 (19.4)
Ikaze	6 (16.7)
Imanzi	4 (11.1)
Umwezi	2 (5.6)
Hirwa	1 (2.8)
**Participant age: (years)**	
18–24	4 (11.1)
25–34	20 (55.5)
35–44	10 (27.8)
45–55	2 (5.6)
**Household size: (number of people)**	
<5	13 (36.1)
5–10	23 (63.9)
**Main cook:**	
Mother	34 (94.4)
Family member	1 (2.8)
Housemaid	1 (2.8)
**Number of children aged <5 years**:	
1	25 (69.4)
2	9 (25.0)
3	2 (5.6)
**Households with electricity access**	36 (100.0)
**Households with chimney present**	36 (100.0)

**Table 2 ijerph-18-05984-t002:** Cooking activities and preferences (*n* = 36).

Cooking Activities	Study Participants *n* (%)
**Type of cookstove used:**	
Single-pot metal charcoal stove (mobile)	35 (97.2)
Multi-pot stone stove (fixed)	1 (2.8)
**Households with multiple single-pot metal charcoal stove**	6 (16.7)
**Biomass fuel used:**	
Charcoal	34 (94.4)
Charcoal and wood	2 (5.6)
**Reason for fuel choice:**	
Cost	33 (91.6)
Tradition	2 (5.6)
Availability	1 (2.8)
**Type of additional lighting fuel used:**	
Plastic	29 (80.6)
Paper	4 (11.1)
Wood	12 (33.3)
Other ^a^	5 (13.9)
**Fuel collection or delivery:**	
Collection by household member	28 (77.8)
Delivery	8 (22.2)
**Number of meals cooked per day:**	
1	19 (52.8)
2	13 (36.1)
3	4 (11.1)
**Location of cooking area:**	
Internal room within main house (living or sleeping area)	4 (11.1)
Separate building attached to the main house (shared wall)	7 (19.4)
Separate building detached from the main house	2 (5.6)
Outside	23 (63.8)
**Approximate time stove in use per day: (hours)**	
>0–2	16 (44.5)
>2–4	12 (33.3)
>4–6	4 (11.1)
>6	4 (11.1)
**Approximate time women spend attending stove per day: (hours)**	
>0–2	16 (44.5)
>2–4	13 (36.1)
>4–6	4 (11.1)
>6	3 (8.3)
**Approximate time child (<5 years) spent near the stove per day: (hours)**	
0	5 (13.9)
>0–2	19 (52.8)
>2–4	8 (22.2)
>4	4 (11.1)

^a^ Other materials used to help light the stove included polystyrene foam, petrol, dry grass and candles.

**Table 3 ijerph-18-05984-t003:** Perceptions of using gas fuel and traditional charcoal stoves (*n* = 36).

Response	Study Participants ^a^ *n* (%)
**Reported perceived benefits of using gas fuel for cooking**	
Cooks food faster	20 (55.6)
Cleaner and improves sanitation	9 (25.0)
Enables cooking indoors	4 (11.1)
Easier to prevent children being near stove	2 (5.6)
Improves the taste of food	2 (5.6)
Reduces/no smoke	16 (44.4)
Health benefits	2 (5.6)
Easier to use and ignite	9 (25.0)
Gas is a cheaper fuel	2 (5.6)
Other ^b^	6 (16.7)
**Reported concerns about using gas fuel for cooking**	
Worried about cost	16 (44.4)
Safety concerns: risk of fires, explosions, injuries	9 (25.0)
Safety concerns: related to children playing with gas	9 (25.0)
Inconvenient to use for cooking some foods	2 (5.6)
Fuel is consumed too quickly	2 (5.6)
Concern about maintenance	2 (5.6)
No Concerns	10 (27.8)
**Maximum monthly cost willing to pay for gas installation? (RWF)**	
0	8 (22.2)
1–1500	10 (27.8)
>1500–3000	12 (33.3)
>3000–4500	3 (8.3)
>4500–6000	2 (5.6)
>6000	1 (2.8)
**Common perceptions of traditional charcoal cookstoves:**	
Associated with disease and poor health	10 (27.8)
Associated with burns and injuries	6 (16.7)
Releases lots of smoke	6 (16.7)
Charcoal is very dirty to cook with	6 (16.7)
Traditional stove breaks easily	7 (19.4)
Charcoal makes food and boiled water taste of smoke	5 (13.9)
Charcoal is slow and time consuming to cook with	3 (8.3)

^a^ Frequencies and percentages do not add up to the total number of study participants (*n* = 36, 100%) because participants could provide multiple responses for a single question. ^b^ Other perceived benefits: no heat from gas stove, association with improved economic status, gas is mobile, gas stove is made from strong materials and is not easily damaged, gas stove takes up less space.

## Data Availability

The data presented in this study are available on request from the corresponding author.

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
