# Peer review of "Investigating Cooking Activity Patterns and Perceptions of Air Quality Interventions among Women in Urban Rwanda"

_ijerph, 2021, doi:10.3390/ijerph18115984_

Round 1

Reviewer 1 Report

The authors of the article "Investigating cooking activity patterns and perceptions of air quality interventions among women in urban Rwanda" present a well-structured and interesting study, concerning the use of traditional stoves and the perception of their impact on health.

The "semi-structured questionnaire" was answered by a limited number of subjects, but this limitation (as well as others that appear in the manuscript) was addressed in subchapter 4.4.

The article shows some pertinent conclusions that are well supported by the data obtained and the discussion that follows. The questionnaire, on the other hand, allowed information about the health risks arising from exposure to HAP from charcoal cooking smoke to spread among the participants, keeping them more aware of its dangers.

Author Response

We thank the reviewer for their positive insightful comments and interest in our novel study. We confirm that a minor spell-check has been applied to the revised manuscript. 

Reviewer 2 Report

The main scope of the article is to present the cooking activity patterns in Rwanda through the conduction of a survey to the local population of Kigali. The study is very interesting given the fact that the administration and the filling of the questionnaire included several obstacles (e.g. language).

Generally the manuscript is well arranged in four sections. A comprehensive literature review is provided and the outcomes  are satisfactorily analyzed in a clear and explanatory manner.

However, some minor corrections mentioned below are either needed or required, in order to correct and improve the paper to be accepted.

1) In table 1 please mentioned the demographic variable that these percentages stand for (e.g. population?).

2) line 185: Please write the percentage as "Sixty four" since it is at the beginning of a sentence. The same for the percentage in line 484

3) If possible the authors should use a scientific tool for the development of the figures 4 and 5.

4) In table 3 please clarify the reason why the responses' number is not the same for all the questions. For instance, at the "reported perceived benefits of using gas fuel for fuel" the responses are 73 while for the "reported concerns about using gas fuel for cooking" there are only 50 responses.

Author Response

  • In table 1 please mentioned the demographic variable that these percentages stand for (e.g. population?).

We confirm these percentages (and associated frequencies) relate to proportions of study participants for each respective category (e.g., 100% denotes all participants). To provide greater clarity and ensure consistency in the revised manuscript we now include ‘Study participants’ in the header rows of Tables 1-3 respectively and the total study sample size (N=36) in the respective table legends.  

  • line 185: Please write the percentage as "Sixty four" since it is at the beginning of a sentence. The same for the percentage in line 484

We have amended the specific percentage numerals to text within the revised manuscript (line 185) and now include ‘Overall’ at the beginning of the aforementioned sentence (line 484).   

  • If possible the authors should use a scientific tool for the development of the figures 4 and 5.

We include revised versions of figures 4 and 5 produced using Stata Statistical software (please see attached). We have amended the methods accordingly and include an additional reference for Stata in the revised manuscript.[i]  

  • In table 3 please clarify the reason why the responses' number is not the same for all the questions. For instance, at the "reported perceived benefits of using gas fuel for fuel" the responses are 73 while for the "reported concerns about using gas fuel for cooking" there are only 50 responses.

We thank the reviewer for inspecting the frequencies reported in Table 3. We confirm that the reason for differences in total frequencies between categories is that study participants were able to select multiple categories within each response (or provide a free text response). Therefore, the cumulative frequency (within each response category) is not expected to be equal to and may exceed the number of study participants (total N=36). To provide greater clarity we include an additional footnote for Table 3 in the revised manuscript (lines 330-332).

[i] StataCorp, 2019. Stata Statistical Software: Release 16. College Station, TX: StataCorp LLC

Reviewer 3 Report

As a reviewer I have the following remarks.

  1. Line 46: “which is greater than any other major modifiable risk factor, including unsafe 46 water and child malnutrition [4]” – is it correct? Better/safer to say “among”. These factors vary by countries and we really need some specifications.
  2. Line 79: “(498.66 people per km2)” – we don’t need .66 , better 498.7.
  3. Line 99: “between January to March 2018.” – one sentence on weather conditions in this period will be good. If rai/cold probably stoves/wood start to be more issue?
  4. Line 170: “plastic was used to aid lighting the stove” – it will be good to know the amount of plastic used –as it creates very toxic fume.
  5. Fig 3. We have cut of the bottom (put Y-axis, say -0.25) and an empty area above (say put Y-axis 10.25).
  6. The paper is well written and presented.
  7. Thank you

Author Response

  • Line 46: “which is greater than any other major modifiable risk factor, including unsafe 46 water and child malnutrition [4]” – is it correct? Better/safer to say “among”. These factors vary by countries and we really need some specifications.

We thank the reviewer for this insightful observation. We confirm that this statement refers to the estimated premature mortality burden for five selected major risk factors in Africa[i] using Global Burden of Disease Study data (2013)[ii], extracted from the Institute for Health Metrics and Evaluation (2015).[iii] We acknowledge there is variation in the relative contribution of household air pollution to overall premature mortality by continent (and country) and have therefore amended this sentence to “which is greater than any other major modifiable risk factor in Africa’ (line 46).

  • Line 79: “(498.66 people per km2)” – we don’t need .66 , better 498.7.

We have amended this figure to 498.7 in the revised manuscript (line 79).

  • Line 99: “between January to March 2018.” – one sentence on weather conditions in this period will be good. If rai/cold probably stoves/wood start to be more issue?

We thank the reviewer for highlighting the importance of prevailing weather conditions which may influence cooking practices. We provide clarification that study fieldwork was undertaken within the ‘short’ dry season in Rwanda (lines 99-100).

  • Line 170: “plastic was used to aid lighting the stove” – it will be good to know the amount of plastic used –as it creates very toxic fume.

We agree with the reviewer that detailed characterisation of all fuels used for stove lighting are relevant for assessment of household air pollution exposure, however this was not the focus of this present study which seeks to characterise cooking activities, preferences and practices. Therefore, study fieldworkers were not trained to obtain or record observations regarding the quantity or volume of specific fuels used to light the cooking stove. We include an additional point on this topic within the discussion section as this finding relevant to future research in this context (lines 380-383).  

  • Fig 3. We have cut of the bottom (put Y-axis, say -0.25) and an empty area above (say put Y-axis 10.25).

We thank the reviewer for noting this discrepancy. We have reproduced Figure 3 in the revised manuscript using Stata software. Due to differences in the outputs provided by Stata for the respective plot we have also  updated the associated text accordingly.  

  • The paper is well written and presented.

We thank the reviewer for their positive comments regarding our manuscript.

  • Thank you

[i] Roy R. The cost of air pollution in Africa. OECD Development Centre Working Papers, No. 333, OECD Publishing, Paris, 2016. Available online: https://doi.org/10.1787/5jlqzq77x6f8-en

[ii] Global, regional and national comparative risk assessment of 79 behavioural, environmental and occupational, and metabolic risks or clusters of risks in 188 countries, 1990-2013. A systematic analysis for the Global Burden of Diseases Study 2013. Lancet 2015;386:2287-323 doi: 10.1016/S0140-6736(15)00128-2  

[iii]Viz Hub GBD Compare. Institute for Health Metrics and Evaluation, University of Washington, Seattle, Available at: http://vizhub.healthdata.org/gbd-compare/